# Relaxation Model of the Relations between the Elastic Modulus and Thermal Expansivity of Thermosetting Polymers and FRPs

**DOI:** 10.3390/polym15030699

**Published:** 2023-01-30

**Authors:** Alexander Korolev, Maxim Mishnev, Dmitrii Ulrikh, Alexander Zadorin

**Affiliations:** 1Department of Building Construction and Structures, South Ural State University, 454080 Chelyabinsk, Russia; 2Department of Town Planning, Engineering Systems, and Networks, South Ural State University, 454080 Chelyabinsk, Russia

**Keywords:** polymers, polymer composites, glass-fiber-reinforced plastics, elasticity, modulus of elasticity, thermal expansivity, coefficient of thermal expansion, relaxation, thermo-relaxation

## Abstract

This research was completed in the development of studies devoted to relations between the elastic modulus (MoE) and thermal expansivity (CTe) of different materials. This study, based on experimental data, confirmed the models of the relations between MoE and CTe under normal and heating temperatures for thermosetting epoxy polymers and glass-fiber FRPs in two variants (unfilled and filled by mineral additives), after the usual glassing and prolonged thermal conditioning (thermo-relaxation). The experiment was based on dilatometric and elastic deformation testing. Two models of MoE/CTe were tested: Barker’s model and our authors relaxation model (MoE = f(CTe)), which is based on previous modelling of the non-linearity of the physical properties of polymers’ supramolecular structures. The result show that the models’ constants depend on composition; Barker’s model is applicable only to polymers with satisfying agreement degrees in the range 10–20%; our model is applicable to polymers and FRPs with satisfying agreement degrees in the range of 6–18%.

## 1. Introduction

Fiber-reinforced plastics are popular and effective modern materials with preferential properties, such as low weight and high strength (especially tensile), excellent corrosion resistance, fatigue resistance, and creep resistance. However, in the case of thermosetting polymers and composites under heating, these properties change sharply and dramatically even under temperatures greater than 50 °C. The prediction of elastic properties of various composite structures, including polymers, under normal and heating temperatures is critical in the design of structures, especially those subjected to transient thermomechanical actions. In structures such as gas and smoke chimneys and pipes for hot liquid transportation, different FRP joints in volume structures heating at temperatures higher than 40–50 °C influence the simultaneous non-linear changing of the modulus of elasticity (MoE) and the coefficient of thermal expansion (CTe). Thermal expansion (TE) and elasticity are the main properties that determine the deformability of the composite and the significant factors of polymers and FRP thermal stresses developing in real structures during exploitation [1,2,3,4,5,6]. Thus, the adequate relation between thermal expansibility and elasticity of materials fuels the decisions of many design tasks. The basic parameter of thermal expansivity is the CTE, and the basic parameter of elasticity is the MoE. Several studies have demonstrated the approximation of relations between MoE and CTE for different materials, including metals, minerals, glasses, polymers, and plastics. The most famous research in this area is the study of R.E. Barker Jr [7].

In Barker’s study, a common study into TE-MoE relation was completed with the following results [7]:The groups of absolutely different materials with approximated TE-MoE relation were determined, and glassed materials, including polymers and FRP, were in this number.The approximated CTE (∝)-MoE (*E*) relation under normal temperature was suggested with the following formula:
(1)$$E\propto^{2}=150\frac{\mathrm{dyn}}{{\mathrm{cm}}^{2}}\times^{0}K^{2}.$$

3.A theoretical energy model of the TE-MoE relation is proposed. The theory is based on the determination of bonds’ harmonical/unharmonical vibration frequencies. The main sequence rule was described by the following function with factors of heat capacity (C) and temperature (T):


(2)
$$C_{p}-C_{v}=\propto^{2}E_{T}T$$


Analyzing these scientific results and taking into account the properties of polymers and FRPs, the following developing questions appear:What is the accuracy of the approximated CTE-MoE relation for thermosetting polymers and FRPs? Research data includes a wide variety of approximations.Is the approximated CTE-MoE relation dependent on temperature? The fact relation (1) between CTE and MoE does not depend on temperature, but the Barker’s theory model (2) has the temperature factor.Could the CTE-MoE relation model be realized based on the physical properties of the polymer’s supramolecular structure? The energy model is very difficult to apply to the application and practical prediction of composites’ elasticity or expansion.

Few studies are concerned with the relationship between the thermal expansivity and elasticity of polymers and plastics. Most of them discover this problem in the research area of stress relaxation under thermal heating [8,9,10,11,12].

Thermal expansion is typical of glassed polymers and materials characterized by CTE, but much research points to the relationship between CTE and heating temperature. Experimental studies of the polymers’ CTE temperature dependence (including epoxy polymers) presented that the temperature expansion depends significantly on not only the temperature range [13,14,15,16,17,18,19], but also on the rate of temperature change [20]. The presence of static stresses during cyclic tests (heating–cooling) is also noted, which leads to a hysteresis effect on the relationship curve between the deformations and temperature. 

In most cases, these phenomena described by the polymer molecule’s confirmation mechanisms resulted in 3D molecule replacement and distribution changes that influenced CTE [21,22,23,24]. Therefore, this confirmation mechanism results in simultaneous CTE and MoE non-linear changing due to stress relaxation and expansivity compensation. According to research on the elasticity of thermosetting polymers under heating, the non-linear decrease in elasticity is a result of the following processes:Polymer molecules thermo-expanding with their morphology and pack changes [25,26].Increased flexibility and torsion of inter-molecular bonds under heating [27,28].

Thus, the relation between CTE and MoE can be realized as the MoE function of CTE factor: MoE = f(CTE), where elasticity is due to thermal expansion. 

Our previous modeling of polymers’ and FRPs’ supramolecular structures [29,30] determined non-linear models of MoE and CTE of polymers and FRPs under heating. The modeling of one-phase supramolecular structure included the spiral as the main element: polymer molecule analogue with non-linear expanding. However, the dilatometric experiment and one-phase model demonstrated an opposite agreement: the spiral model provides a non-linear decrease in CTe, the experiment demonstrated the non-linear decrease in CTe under heating [30]. Only the two-phase model where the spirals are shell joints between rigid and dense crystallized polymer domains agreed with real polymers’ and FRPs’ thermal expansion. The models discover non-linearity as a consequence of the conglomerate character of two-phase supramolecular structures where the domains (phase 1) are dense and rigid centers of polymerization and the spiral shell (phase 2) in the inter-transition zone (ITZ) is a softer joint of peripherical non-dense polymer molecules [30]. The domain works as elastic element with linear properties. The spiral works as a shifting element with the non-linear properties, providing the overall structure properties non-linearity. It determines thermal expansion compensation and stress relaxation during heating, as these properties change after the prolonged heating and thermo-relaxation. Due to that, our model is named as a relaxation supramolecular polymer and FRP model. 

The relaxation model is the last development of our conglomerate supramolecular models of polymers and FRPs that agree with real elastic and thermal deformations under heating. Due to the included relaxation mechanism, the idea appeared that supramolecular thermal expansion and stress relaxation mechanisms are related. Thermal expansion influences the compensation of deformation in ITZ, leading to stress relaxation and MoE decreasing. Additionally, this is the way to the function MoE = f(CTe). It requires the development of the model, designing it under the tensile stress–strain condition.

Modeling the thermal expansion of polymer composites and hybrids is more complex and has to take into account the influence of all components on summary results of stress and expansion. It is important that the non-linearity of the MoE and CTE of polymers and FRPs under heating continues before achieving the glass transition temperature (Tg). After Tg, polymers lose their elasticity and CTE has a linear temperature dependence. So, the Tg is the breaking point of thermosetting polymers’ and FRPs’ deformability models.

All this consideration led to the research aim to develop the model of the relation between the MoE and CTE of thermosetting polymers and PRPs on the basis of the supramolecular structures’ parameters. Accordingly, the following scientific tasks were formulated:To repeat the one-party samples’ experimental research of the MoE and CTE of thermosetted and thermo-relaxed filled and unfilled polymers and FRPs under heating including Tg to determine the accuracy of the approximated CTE-MoE relation.To determinate the approximated CTE-MoE relations’ dependence on temperature and their universality limits.To realize the supramolecular relaxation model in tensile stress–strain condition as a function MoE = f(CTE) of the CTE-MoE relation of thermosetting polymers and FRPs, using previous experience in the supramolecular modeling of composite elasticity and TE.

## 2. Materials and Methods

### 2.1. Materials

The materials used in this study were glassed polymers made of epoxy, phenolic, epoxy-phenolic resins, and fiberglass plastics made of epoxy and epoxy-phenolic resins and structural glass fabrics EZ-200 and T-23 (Table 1). In this study, new samples of compositions, repeated (1, 2) and non-repeated from previous studies, were used, allowing for the simultaneous MoE and CTe testing of the same series’ samples.

Epoxy binder (EP) for fiberglass plastic was created based on epoxy resin KER 828 (South Korea), which is an analog of the Russian resin ED-20, isomethyltetrahydrophthalic anhydride (ISOMTHFA) was used as a hardener, 2,4,6-tris-(dimethylaminomethyl)-phenol, produced under the brand name Alkophen, was used as a curing gas pedal. The weight ratio of the ES components is as follows: KER 828-52.5 %, IZOMTGFA-44.5 %, Alcophene-3 %. The components described below were used to produce the binders:Epoxy resin KER 828 with the following main characteristics: epoxy group content (EGC) 5308 mmol/kg, epoxide equivalent weight (EEW) 188.5 g/eq, viscosity at 25 °C 12.7 Pa.s, HCl 116 mg/kg, and total chlorine 1011 mg/kg. Manufacturer: KUMHO P&B Chemicals, Gwangju, South Korea.Hardener for epoxy resin methyl tetrahydrophthalic anhydride with the following main characteristics: viscosity at 25 °C 63 Pa.s, anhydride content 42.4%, volatile fraction content 0.55%, and free acid 0.1%. Manufacturer: ASAMBLY Chemicals company Ltd., Nanjing, China.Alkofen (epoxy resin curing accelerator) with the following main characteristics: viscosity at 25 °C 150 Pa.s, molecular formula C15H27N3O, molecular weight 265, and amine value 600 mg KOH/g. Manufacturer: Epital JSC, Moscow, Russian Federation.

The components were mixed in the above proportions at room temperature of about 25 °C. Mixing to a homogeneous consistency was carried out mechanically with an electric drill with a mixing attachment.

Composites were produced by the introduction of glass fiber and mineral additives.

Glass fabric T-23 was produced in accordance with Russian standard GOST 19170-2001 and has the following characteristics:-Thickness, 0.27 + 0.01/−0.02 mm;-Surface density, 260 + 25/−25 g/m^2^;-Number of yarns per 1 cm of fabric on the basis 12 ± 1;-Number of yarns per 1 cm of fabric on the weft 8 ± 1;-Weave—plain;-Oiling agent—aminosilane.

The dry fly ash was chosen as a mineral additive, providing the glass structure with all materials used in Barker’s (the most effective for glassed materials) and our own models testing. A mineral additive was the dry fly ash (FA) from Refta electric station with square surface 4500 cm^2^/g; the dominating oxides are SiO_2_ (>70%), Al_2_O_3_ (>12%), Na_2_O+K_2_O (>6%). The fly ash proportion (Table 1) was determined by the technological optimum that provided the filling of glass fiber in the FRP shells’ filament winding formation.

Samples of fiberglass plastic and composites were produced in the form of plates of 15 × 15 cm. Cut sheets of glass fabric EZ-200 were calcined at 300 °C to remove the paraffin oiling agent immediately before impregnation with the binder. Glass fabric T-23 was not calcined. In total, the samples had 10 layers of glass fabric laid according to the scheme 0/90 (base/weft). 

Glass-reinforced plastic and composite specimens were cured at 120 °C for 20 min in silicone molds while being loaded through Teflon-coated metal plates at a pressure of about 0.22 kPa. The cured specimens were then kept at 150 °C for 12 h. After that, beam samples were cut from the plates in the direction of the main axes of orthotropy, which were considered in this work.

### 2.2. Methods

#### 2.2.1. Long Heat Treatment (Thermal Aging)

After curing, some of the fiberglass samples were exposed to prolonged exposure at elevated temperatures, while the control series was stored under normal conditions. The long-term curing (hereinafter referred to simply as “curing”) of the samples at elevated temperatures was performed according to the following program: 168 h (one week) at 160 °C, 168 h at 190 °C, 168 h at 220 °C. After the heat treatment, the samples were cooled at a rate of about 1 °C per minute to 50 °C, removed from the laboratory oven, weighed, and then weighed and tested for three-point bending at temperatures from 25 to 180 °C.

The long heat treatment has a wide application background, because it reflects the real chimneys’ and pipes’ shells under exploiting heating (under 60–300°C depending on the mission) MoE and CTe change, which leads to a non-linear change in the structure’s deformability during the long exploiting heating. Consideration of this non-linearity can provide an improvement in the design of FRP structures. 

#### 2.2.2. Dilatometric Investigation 

Dilatometric investigations of the polymer and FRP samples were carried out with dilatometer Netzsch DIL 402 C (Figure 1). Investigation determined the CTE of solid materials analogically to previous research [28], using new series of samples allowed tests of the elasticity and TE for the same series.

Netzsch DIL 402 C technical characteristics:-Temperature range: 20–1500 °C;-Colding and heating intensity: 0.01 °C/min–50 K/min (5 K/min in experiment);-Etalon: Al_2_O_3_;-Linear range: 500 mcm;Sample length *l*: max. 28 mm;-Sample diameter: max. 12 mm;Expanding Δ*l* accuracy: 0.125 nm;-Atmosphere: inertial dynamic argon with gas flowing controller.

At Figure 1, the samples’ installation in the dilatometer’s camera is shown. After the installation, the heating and thermal deformations of samples were controlled, thermal expanding automatically was calculated as a relation $\frac{dl}{l}$ under heating temperature, resulting in “temperature-relative deformation” curves presented in the experimental part.

#### 2.2.3. Investigation of Elasticity Modulus under Heating 

Polymer samples were tested for three-point bending on a Tinius Olsen h100ku test machine (Switzerland) in a specially made small-sized chamber that provides heating and maintains the temperature up to 300 °C. Three-point bending tests were carried out according to the Russian standard GOST R 56810-2015. The bending test is preferable in comparison with the tensile test for determining the modulus of elasticity at elevated temperatures due to a number of reasons. During the tensile test, the ends of the sample are in a pinched state in the clamps of the testing machine. Therefore, when heated, temperature stresses appear, which are difficult to separate from the applied mechanical stresses, occurred in the sample, in the determined modulus of elasticity. During the bending test, the sample can extend more freely when heated, because its ends are affected only by friction against the supports, so temperature stresses should not in this case introduce a significant error in the result of determining the modulus of elasticity.

According to the passport data, the load measurement accuracy of the Tinius Olsen h100ku machine is ±0.5% in the range from 0.2 to 100% of the allowable load of the installed force sensor (100 kN). The crosshead has a resolution of 0.001 mm with an accuracy of 0.01 mm. To eliminate the influence of machine stickiness, the displacement of the specimen center point under load was also controlled by a mechanical watch-type indicator mounted under the specimen. The difference in displacement readings on the crosshead and the dial indicator did not exceed 2%. The specimens were tested at a span of 70 mm (Figure 2).

## 3. Results

### 3.1. Experiment Results

On the basis of dilatometric and MoE under heating tests of the glassed thermosetting polymers, filled composites, and FRP samples, the thermal expanding and MoE curves dependent on heating temperature (to 200 °C) were calculated and presented in Figure 3, Figure 4, Figure 5 and Figure 6. The dependences include the clear epoxy (EP), filled epoxy composite (EP+FA), FRP (EP+T23), and filled epoxy FRP variants (EP+T23+FA) before and after thermo-relaxation (TR). Testing of MoE and CTe included only new samples of one series, providing the experiment with clearance and adequateness of the modelling of MoE/CTe relations.

These results have repeated and confirmed previous research [29,30] and basic conclusions:The polymers’ glassing temperature Tg can be determined, using a dilatation method, by the breaking point of the thermal expansion (TE) curve where non-linear TE with increase in CTE becomes linear with constant CTE.Under heating, after Tg, polymers lose their elasticity absolutely. The thermo-relaxed polymers, filled polymers, and FRPs, after Tg, lose elasticity sharply but can keep it, depending on the conditions, until relatively high temperatures (160–200 °C).The long heat treatment (thermo-relaxation (TR)) significantly changes all the properties of polymers and polymer composites: the Tg grows to 30–40%, the TE decreases, the temperature of the coworking of components in composites and their elasticity keeps growing several times. After the long heat treatment, MoE at normal temperature recovers, but 3–5% less than before heating; however, MoE under repeating heating grows at high temperatures several times.The TE curves have an inversive character in relation with MoE under heating curves. Generally, growth of CTe correlates with MoE decrease.

### 3.2. Supramolecular Relaxation Model of CTE-MoE Relation of Thermosetting Polymers and FRPs

The previous research proved the prediction compensational TE model of thermosetting polymers to be adequate [30]. The main specification of the compensational model is that in two-phase structure rigid elements (spherulitic domains) with linear TE are joined by softer elements (ITZ as the polymer macromolecules’ spiral shell) with shifting bonds and non-linear TE. The Tg is the extremum of the TE non-linear function. In Figure 7, the previous spiral and conglomerate 2D models were combined with a tensile stress–strain model where stresses and deformations influence the deformation of spirals, changing the spiral’s branch angle and separate forces and deformations by the X and Y axes. All deformations of domains influence the respective deformations of the shell. The tensile domain’s expansion by the Y-axis influences the spiral shell’s compression by the Y-axis and the shell’s tensile stress by the X-axis. It leads to the spiral’s branch angle β decreases, and a respective decrease (relaxation) in tensile force in the branch $F\times sin\beta_{t}$ and a growth of the tensile cross-section’s square $S_{t}$. It results in the total tensile stress under heating relaxation, the growth of the shift modulus, and a decrease in the tensile MoE. The maximum expansibility is constant because of the spiral geometry—the spiral’s maximum expansibility by Y and compressibility by X. 

Thus, on the basis of the supramolecular polymer’s model analysis, the following formulated hypothesis appeared: the total polymer’s expansion under heating leads to the ITZ spiral shell’s tensile stress along the domain surface (X-axis) and to the stress relaxation perpendicular to the domain surface (Y-axis) due to the growth of the stressed cross-section and to the redistribution of tensile forces in spiral branches by the X- and Y-axis. 

For math modeling, the supramolecular model (Figure 7) was modified by the tensile forces F distributed in the polymer molecules’ branches, depending on the branch’s angle β. The following math model is developed from MoE under the definite heating temperature function:(3)$$E_{t}=\frac{\sigma_{t}}{\varepsilon_{t}}$$

$\sigma_{t}$ is the stress in the spiral branch.

$\varepsilon_{t}$ is the structure’s relative tensile strain/deformation under stress.

The following equations are the author’s development:

The spiral’s relative tensile strain/deformation $\varepsilon_{t}$ under tensile stress and heating to temperature t by the Y-axis is equal to the spiral branch’s relative deformation $\varepsilon_{0}$
(4)$$\varepsilon_{t}=\frac{\Delta y}{y}=\frac{\Delta l\times cos\beta }{l\times cos\beta }=\frac{\Delta l}{l}=\varepsilon_{0}$$

l is the length of the spiral polymer molecule’s branch (Figure 7, top scheme);

Δl is the expansion of the spiral polymer molecule’s branch (Figure 7, top scheme);

β is the angle of the slope of the spiral polymer molecule’s branch (Figure 7, top scheme).

Decrease in the MoE under heating is related with tensile stress relaxation and a redistribution of tensile and shift stresses; the more heating, X-tensile and Y-compression, the less Y-tensile and more X-shift stresses.
(5)$$E_{t}=\frac{\sigma_{t}}{\varepsilon_{0}}$$

The tensile stress under definite temperature is the relation between tensile force in the spiral branch ($F_{t}$) and total square ($S_{t}$) of the cross section under a definite temperature.
(6)$$\sigma_{t}=\frac{F_{t}}{S_{t}}=\frac{F\times sin\beta_{t}}{b\times l\times cos\beta_{t}}=\sigma_{0}\times tg\beta_{t}$$

$\beta_{t}$ is the angle of the slope of the spiral polymer molecule’s branch under heating (Figure 7, bottom).

So, the MoE under a definite temperature depends on the MoE under normal temperature ($E_{0}$) by the following model:(7)$$E_{t}=E_{0}\times tg\beta_{t}$$

Use Equation (14) of angle $\beta_{t}$ from the previous CTE modelling by compensational model [30].
(8)$$E_{t}=E_{0}\times tg\left[arctg\left(1-\frac{\alpha_{t}}{\alpha_{tg}}\right)\right]=E_{0}\left(1-\frac{\alpha_{t}}{\alpha_{tg}}\right)$$

$\alpha_{t}$—CTE under definite heating temperature;

$\alpha_{tg}$—CTE under glassing temperature Tg.

Thus, the constant of MoE-CTE relation means the basic MoE of the polymer’s supramolecular ITZ shell under a normal temperature.
(9)$$E_{0}=E_{t}\frac{\alpha_{tg}}{\alpha_{tg}-\alpha_{t}}$$

For polymers under a normal temperature of 20 °C
(10)$$E_{0}=E_{20}\frac{\alpha_{tg}}{\alpha_{tg}-\alpha_{20}}$$

That is the searching function MoE = f(CTE) and relaxation model of MoE-CTE relation. Further investigation was devoted to the approbation of Barker’s and our own relaxation models.

### 3.3. Testing of Relations between MoE-CTE Models

The processing of the polymers’ and FRPs’ TE and MoE data included a determination of experimental CTE ($\alpha_{t}$) and MoE ($E_{t}$) values under the same character temperatures from Figure 4, Figure 5, Figure 6 and Figure 7. After that, calculations from Barker’s (1) and our own relaxation (9) models were performed. Test of the models of the MoE-CTe relations concluded in correlation between two experimental data arrays of the MoE and CTe of polymers and FRPs. It was the reason for assessing the degree of agreement between the models using the variation coefficient $C_{v}$. Variation coefficient was determined for every composition by the equation
(11)$$C_{v}=\frac{S_{y}}{\bar{y}}100\%$$

$\bar{y}$—average arithmetical;

$S_{y}=\sqrt{\overset{n}{\underset{i=1}{\sum }}{\left(\bar{y}-y_{i}\right)}^{2}}$ is average square variation.

Finally, for each model the average $C_{v}$ was calculated. Only the parameters of compositions before Tg were considered in the calculation of variation characteristics. It is related with absolute changes in dependences’ character and mechanism after Tg. The results of the calculation of the parameters and functions are presented in Table 2. 

The average $C_{v}$ for different models and compositions are presented in Table 3.

On the basis of these results the following conclusions were made:The character of CTe and MoE under heating dependencies presents that a growth of polymer or composite CTe due to the heating or due to the growth of CTes’ of a composition’s additional components correlates with a decrease in MoE, which highlights the multiply functions in math modelling of the MoE-CTe relation.The tested MoE-CTE relation models of definite composition under heating can be considered applicable and adequate under these conditions. The variation coefficients differ in a wide range from 6 to 36%, and the accuracy depends on temperature. The most variation is noticed closer to Tg. In summary, our own model has a higher agreement degree than Barker’s.The tested MoE-CTE relation models do not have universality. Constants of the models depend on the composition type.Authors relaxation model’s constant *E*_0_ characterizes the elasticity of definite polymer or composite. It is always more than the fact MoE of the composition and it suggests that this constant is the potential maximal MoE of the composition under the extreme density of its supramolecular structure.The average *C_v_* results lead to the conclusion that in conditions under the variation of no more than 20%, Barker’s model is adequate only for polymers and filled polymers, our own relaxation model has the most accuracy and is more applicable to polymers and composites overall.

## 4. Conclusions

In the results of the presented study all proposed questions were answered:The tested MoE-CTe relation models are applicable and adequate under the conditions.These models are adequate under heating.The relaxation model on the basis of a supramolecular structure’s properties was realized and comparatively more accurate.

The dilatometric and MoE tests under the heating of thermosetting polymers, filled polymers, and FRPs were performed after normal thermosetting and prolonged thermo-relaxation. On the basis of the CTE and MoE parameters, the supramolecular relaxation model of MoE-CTE relation was realized, and our and Barker’s models were tested and compared.

The relaxation model has a compensational mechanism, realized due to the conglomerate two-phase structure of rigid domains and softer ITZ spiral shells with the effect of thermal expansion compensation and tensile stress relaxation under heating. The math function MoE = f(CTE) was developed. The constant of the relaxation model (9) is the potential maximum MoE of polymer or polymer composites in normal conditions with a maximal possible density of the polymer molecules’ pack in the supramolecular structure. Test and calculation data (Table 2) present that the potential elasticity of the structures of polymers and polymer composites are 1.5 to 2 times more than fact. It discovers an additive potency to find the ways of increasing the elasticity of polymers and FRPs under normal and high temperatures, modifying their supramolecular structures.

Finally, the relaxation model of MoE-CTE relation has been designed and proved and may be applied to operative values of thermal expansion by MoE parameters or vice versa: for MoE prediction by the parameters of thermal expansion TE, in the design of polymer composite structures, and in calculations of the thermal and thermo-mechanical stresses and deformations.

## Figures and Tables

**Figure 1 polymers-15-00699-f001:**
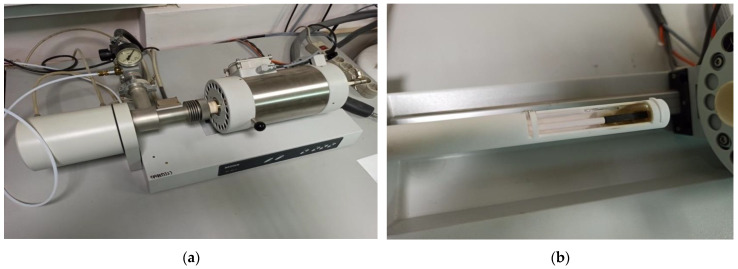
Dilatometer Netzsch DIL 402 C (**a**) in process; (**b**) with sample (brown).

**Figure 2 polymers-15-00699-f002:**
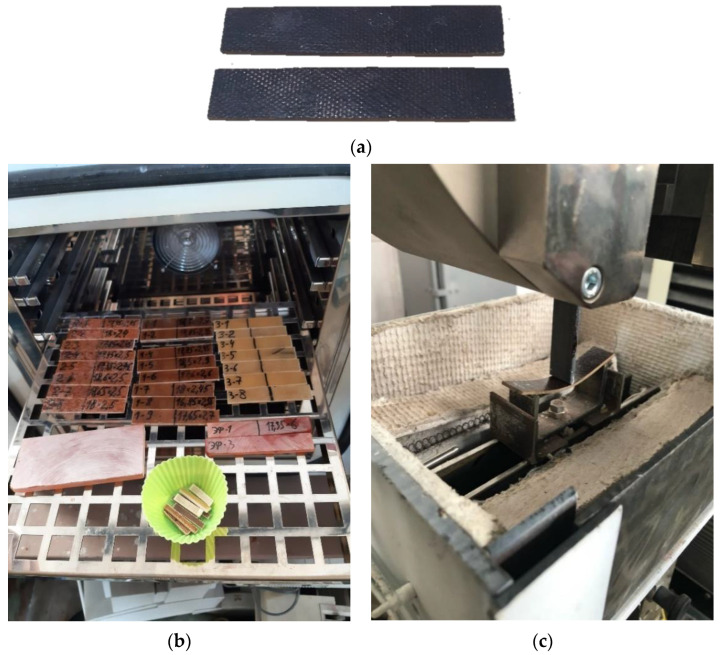
Appearance of fiberglass specimens after prolonged conditioning at elevated temperatures (**a**), specimens prepared for exposure (**b**), and three-point bending process at room temperature (**c**).

**Figure 3 polymers-15-00699-f003:**
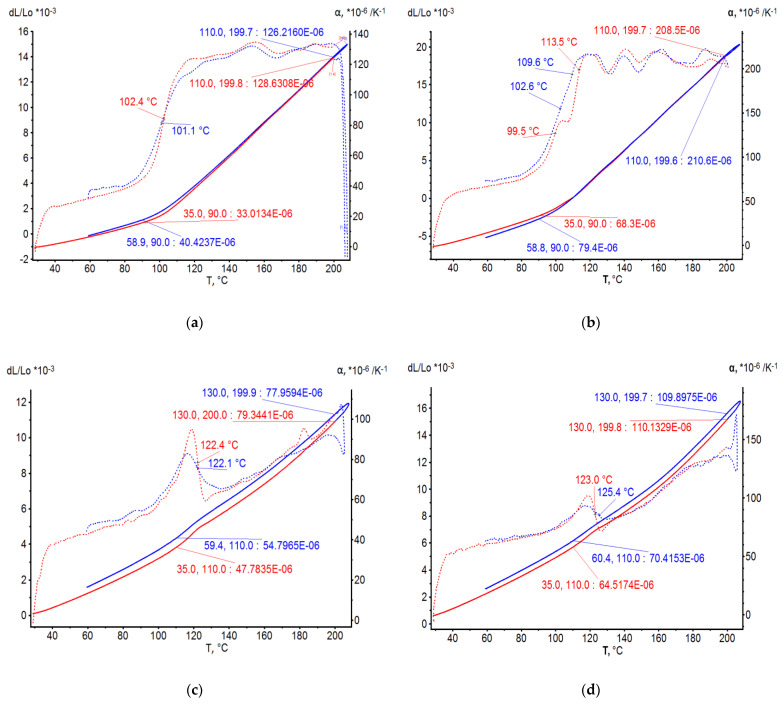
Polymers’ TE curves: (**a**) EP, (**b**) EP+FA, (**c**) EP after T-relax, (**d**) EP+FA after T-relax.

**Figure 4 polymers-15-00699-f004:**
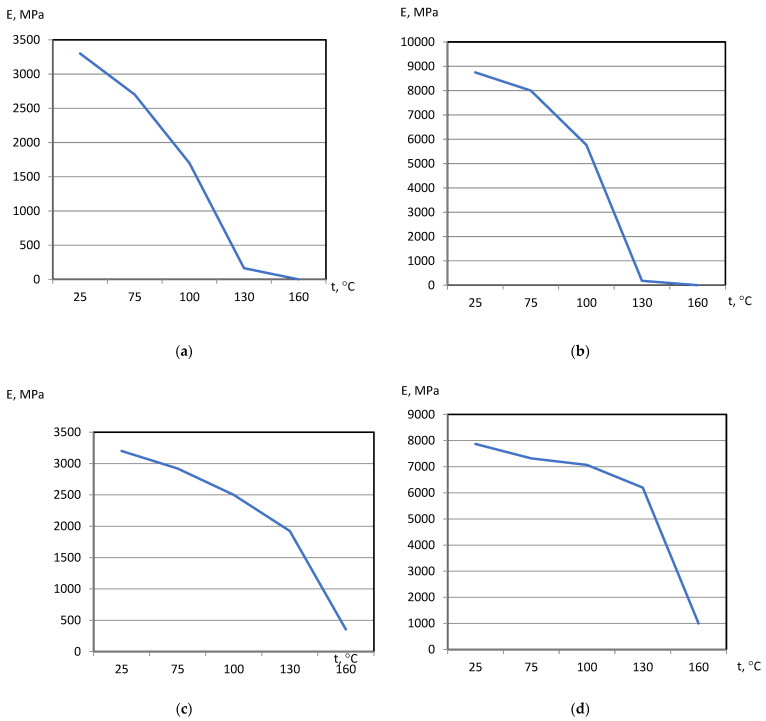
Polymers’ MoE under heating curves: (**a**) EP, (**b**) EP+FA, (**c**) EP after TR, (**d**) EP+FA after TR.

**Figure 5 polymers-15-00699-f005:**
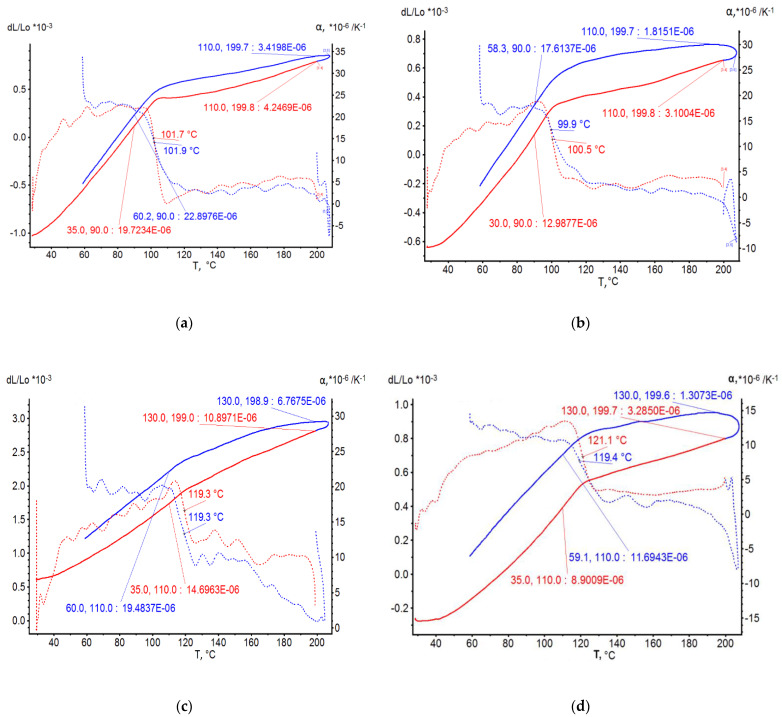
FRPs’ TE curves: (**a**) EP+T23, (**b**) EP+T23+FA, (**c**) EP+T23 after T-relax, (**d**) EP+T23+FA after TR.

**Figure 6 polymers-15-00699-f006:**
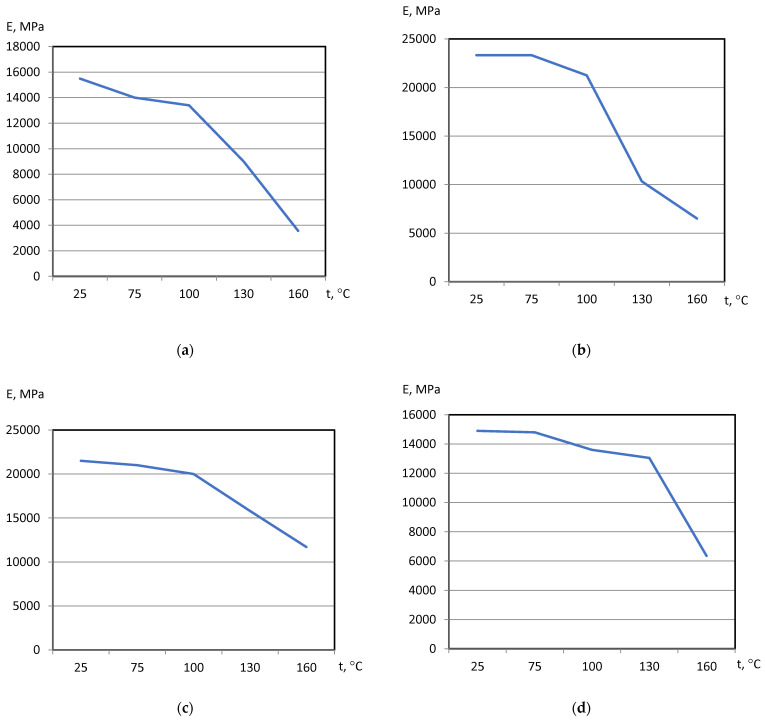
FRPs’ MoE under heating curves: FRPs’ TE curves: (**a**) EP+T23, (**b**) EP+T23+FA, (**c**) EP+T23 after T-relax, (**d**) EP+T23+FA after TR.

**Figure 7 polymers-15-00699-f007:**
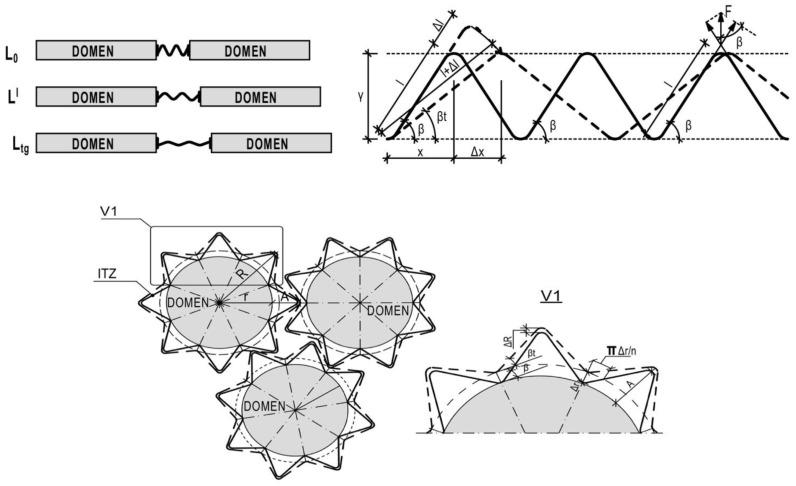
Two-phase relaxation model of polymer expanding under heating.

**Table 1 polymers-15-00699-t001:** Types of composites and FRPs investigated.

№	Composite	Name	Dilatometry	Modulus of Elasticity
1	Epoxy binder	EP	+	+
2	Epoxy binder + glass fabric T23	EP+T23	+	+
3	Epoxy binder 70% + fly ash 30%	EP+FA	+	+
4	Epoxy binder 70% + fly ash 30% + glass fabric T23	EP+T23+FA	+	+

**Table 2 polymers-15-00699-t002:** Results of calculation of MoE-CTE relation models’ parameters and agreement degrees.

Composition	Parameter/Function	T, °C						Average	C_v_, %
		20	75	100	130	160	*α_tg_*		
EP	$E_{t}$, MPa	3300	2700	1700	164	0	-	-	-
	$\alpha_{t}$, K^−1^, 10^6^	21.0	26.0	38.0	57.0	125.9	128.0	-	-
	$E\propto^{2}$, MPa^.^K^2^	1,455,300	1,825,200	2,454,800	532,836	0	-	1,911,767	37
	$E_{0}=E_{t}\frac{\alpha_{tg}}{\alpha_{tg}-\alpha_{t}}$, MPa	3948	3388	2418	167	0	-	3251	34
EP/TR	$E_{t}$, MPa	3200	2920	2500	1924	354	-	-	-
	$\alpha_{t}$, K^−1^, 10^6^	36.0	40.0	42.6	50.4	54.0	79.3	-	-
	$E\propto^{2}$, MPa^.^K^2^	4,147,200	4,672,000	4,536,900	4,887,267	1,032,264	-	4,560,841	12
	$E_{0}=E_{t}\frac{\alpha_{tg}}{\alpha_{tg}-\alpha_{t}}$, MPa	5861	5892	5402	5279	360	-	5608	10
EP+FA	$E_{t}$, MPa	8750	8000	5760	180	0	-	-	-
	$\alpha_{t}$, K^−1^, 10^6^	55.0	60.0	73.0	100.0	125.9	208.0	-	-
	$E\propto^{2}$, MPa^.^K^2^	26,468,750	28,800,000	30,695,040	1,800,000	0	-	28,654,597	10
	$E_{0}=E_{t}\frac{\alpha_{tg}}{\alpha_{tg}-\alpha_{t}}$, MPa	11,895	11,243	8875	347	0	-	10,671	21
EP+FA/TR	$E_{t}$, MPa	7870	7320	7070	6200	1000	-	-	-
	$\alpha_{t}$, K^−1^, 10^6^	50.0	54.0	60.0	63.8	70.0	110.0	-	-
	$E\propto^{2}$, MPa^.^K^2^	19,675,000	21,345,120	25,452,000	25,236,728	4,900,000	-	22,927,212	22
	$E_{0}=E_{t}\frac{\alpha_{tg}}{\alpha_{tg}-\alpha_{t}}$, MPa	14,428	14,379	15,554	14,762	1009	-	14,781	6
EP+T23	$E_{t}$, MPa	15,490	14,000	13,400	9000	3560	-	-	-
	$\alpha_{t}$, K^−1^, 10^6^	10.0	11.0	12.5	10.2	8.30	22.0	-	-
	$E\propto^{2}$, MPa^.^K^2^	1,549,000	1,694,000	2,093,750	936,360	245,248	-	1,778,917	22
	$E_{0}=E_{t}\frac{\alpha_{tg}}{\alpha_{tg}-\alpha_{t}}$, MPa	28,398	28,000	31,032	16,780	3562	-	29,143	8
EP+T23/TR	$E_{t}$, MPa	21,500	21,000	20,000	15,800	11,700	-	-	-
	$\alpha_{t}$, K^−1^, 10^6^	9.5	10.0	12.0	13.8	13.3	19.4	-	-
	$E\propto^{2}$, MPa^.^K^2^	1,940,375	2,100,000	2,880,000	3,008,952	2,069,613	-	2,482,331	38
	$E_{0}=E_{t}\frac{\alpha_{tg}}{\alpha_{tg}-\alpha_{t}}$, MPa	42,131	43,340	52,432	54,736	11,707	-	48,160	23
EP+T23+FA	$E_{t}$, MPa	23,330	23,330	21,250	10,330	6500	-	-	-
	$\alpha_{t}$, K^−1^, 10^6^	16.0	17.0	18.0	14.0	11.7	22.9	-	-
	$E\propto^{2}$, MPa^.^K^2^	5,972,480	6,742,370	6,885,000	2,024,680	889,785	-	6,533,283	11
	$E_{0}=E_{t}\frac{\alpha_{tg}}{\alpha_{tg}-\alpha_{t}}$, MPa	77,429	90,552	99,311	10,336	6503	-	89,097	17
EP+T23+FA/TR	$E_{t}$, MPa	14,900	14,800	13,600	13,050	6350	-	-	-
	$\alpha_{t}$, K^−1^, 10^6^	5.5	5.8	6.9	8.0	7.0	11.6	-	-
	$E\propto^{2}$, MPa^.^K^2^	450,725	497,872	647,496	835,200	311,150	-	607,823	49
	$E_{0}=E_{t}\frac{\alpha_{tg}}{\alpha_{tg}-\alpha_{t}}$, MPa	28,334	29,600	33,566	42,050	6353	-	33,388	32

**Table 3 polymers-15-00699-t003:** Results of calculation of models’ average $C_{v}$.

Average *C_v_*%	Polymers and Filled Polymers	FRPs	Total
Barker’s model	20	30	25
Relaxation model	18	20	19

## Data Availability

The data that support the findings of this study are available on request from the corresponding author.

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
