# Peer review of "Relaxation Model of the Relations between the Elastic Modulus and Thermal Expansivity of Thermosetting Polymers and FRPs"

_polymers, 2023, doi:10.3390/polym15030699_

Round 1
Reviewer 1 Report (New Reviewer)
The Relaxation Model of Relation between Elastic Modulus and Thermal Expansivity of Thermosetting Polymers’ and FRPs was developed. The research work in this paper is crucial to promote the application of polymer and composites under normal and heating temperatures. However, the following comments should be responded to make the improvements.
1. Abstract, it is suggested to provide some qualitative and quantitative results or conclusions, and describe the establishing process the model and the agreement degree between the model and experimental results.
2. Introduction, the basic information and application background of FRP should be clarified firstly, for example, light weight and high strength, excellent corrosion resistance, fatigue resistance and creep resistance. In addition, the importance of the long-term service properties of FRP at high temperatures should also be further explained. After the above two points are analyzed clearly, the research topic of this paper on the relationship between thermal expansion coefficient and elastic properties of materials can be derived. Please review the latest research work below to make necessary additions, such as Polymer testing, 2020; 90: 106761. Polymers, 2022, 14(19): 4002. Construction and Building Materials, 2022, 314: 125587.
3. In the last paragraph of the introduction, it is suggested that the authors provide the contributions and innovations of the current research, such as application fields and scenarios of the proposed model.
4. In part 2.1 and table 1, why do you add the fly ash into epoxy resin and composites? Will the addition of this material affect the interfacial bonding of fiber and resin? In addition, how to determine the selected volume fraction or mass fraction of epoxy resin and fly ash?
5. For the long heat treatment, are the selected exposed temperatures simulating high-temperature fire environment or other service environments? Is there any practical application background?
6. For the investigation of elasticity modulus under heating, Is the author planning to use three-point bending test to obtain the bending modulus of materials? Why not conduct the tensile test to obtain the elastic modulus of the material? It is suggested to provide relevant explanations.
7. Is the 3-point bending test conducted after heat exposure? Will the test after exposure lead to the recovery of elastic modulus compared with the test during exposure?
8. Figure 3-6 shows the thermal expanding and MoE under heating curves for different materials. It is suggested to combine the above four figures to further analyze the effect of different materials on thermal expanding and MoE. At present, it is difficult to detect the difference of the thermal properties and MoE of different materials.
9. What is the agreement degree between the model and the experimental results? It is recommended to provide relevant comparison results and analysis.
Author Response
See the attachment

Reviewer 2 Report (New Reviewer)
The title has to be corrected in stead of "expanding" use "expansion".
The mathematical expressions calculating thermal expansivy should be presented.
What is the physical significance of thermal expansivity in a construction material in relation to mechanical performance? Can the results be compared through DMA? The authors should give a suitable justification.
Since this paper is denoted as Part-I, there should be an indication as to what will be the future study in the conclusion section and what will be presented as Part-II?
Why capitalization in the title?
Round 2
Reviewer 1 Report (New Reviewer)
Accepted.
This manuscript is a resubmission of an earlier submission. The following is a list of the peer review reports and author responses from that submission.
Round 1
Reviewer 1 Report
Authors report the relaxation model of relation between 2 elastic modulus and thermal expansivity of 3 thermosetting polymers’ and frps’. Models demonstrated the modulus and thermal expansion of the fiber filled composites. This work is quite interesting and has physical significance. I am recommending the manuscript for publication with following minor comments. What is strain used to model it? Does diameter and length control this modulus and thermal expansion characteristic?
Author Response
|
Dear Reviewer, The authors would like to express their sincere gratitude for the additional comments that have improved the work. These recommendations will also be taken into account in future studies. |
|
|
Ø F1. What is strain used to model it? |
In the study tensile strain was used. It was added in 261-263 line |
|
Sec2. Does diameter and length control this modulus and thermal expansion characteristic? |
FRPs’ samples size influences the MoE results, but we used close sizes of samples for MoE and CTe testing, so we think that we excluded scale factors from relation |

Reviewer 2 Report
In this study, the authors analysed a possible correlation between the elastic modulus (MoE) and thermal expansivity (CTe) of different thermosetting and composite systems. Different models have been compared though the experimental results, performed also in standard and heating conditions.
Following some suggestions:
· -the abstract should be more extended in accordance with the journal format (max 200 words). Materials, methods and results should be described
· - all the acronyms should be explained
· -Terms of equations 1 and 2 should be explained
· - what the dots indicate in Temperature range: "20 … 1500°C" "Colding and heating intensity: 0.01 K/min … 50 K/min" Please use a different symbol
· -What does mean: Linear range: 500 mcr (please use SI units)
· -Please, explain the dilatometric method through images shown in Figure 2.
· -Pay attention to axis E in MPa and temperature T, °C
· -the description of results is too sparse. First of all, figures should be introduced, then, the trend should be deepened and described.
· - Results of dilatometric and mechanical measurements should be divided and explained separately
· -“as adequate []” missing reference
Reviewer 3 Report
Dear Authors,
Your manuscript has not been prepared in a proper way, e.g.:
in Fig. 4, 5, 6, 7. units in Russian (MPa) appear - the language of the journal is English;
the text seems to lack citations, because there are only empty brackets [];
already on the first page of the manuscript, it is obvious how sloppy the text has been prepared if there is a problem with the authors and their affiliations;
there are a huge number of Equations in the article and they are not mentioned in the text - only maybe 1 (Barker) and 9 (Authors);
in the presented text there is a detailed description of materials and methods, there are also conclusions, but there is no detailed analysis and discussion of the results;
The manuscript prepared by you is not suitable for publication and I propose to reject it.
Author Response
|
Dear Reviewer, The authors would like to express their sincere gratitude for the additional comments that have improved the work. Many comments and recommendations will also be taken into account in future studies. |
|
|
1. in Fig. 4, 5, 6, 7. units in Russian (MPa) appear - the language of the journal is English;
|
It was corrected |
|
2. The text seems to lack citations, because there are only empty brackets [];
|
It was corrected |
|
3. Already on the first page of the manuscript, it is obvious how sloppy the text has been prepared if there is a problem with the authors and their affiliations |
We don’t have problems with author’s affiliation, but there were mistakes in preparing of manuscript, not study and results. Thanks for comment |
|
4. There are a huge number of Equations in the article and they are not mentioned in the text - only maybe 1 (Barker) and 9 (Authors)
|
Except Barker’s the all other equations are author’s modelling, we added the comment about this at line 262 |
|
5. In the presented text there is a detailed description of materials and methods, there are also conclusions, but there is no detailed analysis and discussion of the results |
we added some analyze at lines 303-316 |

Reviewer 4 Report
The manuscript entitles" THE RELAXATION MODEL OF RELATION BETWEEN
ELASTIC MODULUS AND THERMAL EXPANSIVITY OF THERMOSETTING POLYMERS’ AND FRPs’" is an excellent journal and can be an excellent addition to the polymer research community. The scientific impact of this work is undeniable, however, the authors should include the following information that will make it easier for the reader of the research community-
1. the authors should include sufficient references for every equation that was used in the manuscript.
2. The background of every equation and the correlations among them are difficult to understand. It is highly recommended to add a supplementary section where readers can understand it properly.
Author Response
|
Dear Reviewer, The authors would like to express their sincere gratitude for the additional comments that have improved the work. The comments and recommendations will also be taken into account in future studies. |
|
|
1. the authors should include sufficient references for every equation that was used in the manuscript. |
Except Barker’s the all other equations are author’s modelling, we added the comment about this at line 262 |
|
2. The background of every equation and the correlations among them are difficult to understand. It is highly recommended to add a supplementary section where readers can understand it properly. |
The basic modelling was made in previous work in citation [28], but we added some discovering sections at lines 258-285 |

Round 2
Reviewer 2 Report
Suggestions have been sufficiently met, and however some aspects have been explained.
Author Response
Dear Reviewer,
The authors would like to express their sincere gratitude for the additional comments that have improved the work. Many comments and recommendations will also be taken into account in future studies.

Reviewer 3 Report
Dear Authors,
I believe that the manuscript you are presenting should not be published.
Korolev, A.; Mishnev, M.; Ulrikh, D.V. Non-Linearity of Thermosetting Polymers’ and GRPs’ Thermal Expanding: Experimental Study and Modeling. Polymers 2022, 14, 4281. https://doi.org/10.3390/ polym14204281
Citations of your earlier work have been added to the work, and as it turned out, several Figures (1, 2, 8) and graphs (Fig. 4-7) were also used in this work. I treat it as plagiarism and unacceptable activity / ethical concerns. Please don't treat an open access journal as an opportunity to publish anything for money.
